

# Trends in dyslipidemia prevalence among Uyghur adults of different genders in China: a retrospective cohort study

Gulinigaer Maimaitituersun[*], Subinuer Jureti[*], Ziyu Yi, Yaqi Zhou, Meng Li, Mengwei Wei, Ziyang Liu, Menglong Jin and Zhenyan Fu

The First Affiliated Hospital, Xinjiang Medical University, Urumqi, Xinjiang, China
Xinjiang Medical University, Wulumuqi, China
[*] These authors contributed equally to this work.

Corresponding author
Zhenyan Fu, fuzhenyan316@126.com

## ABSTRACT

**Background.** To analyze the prevalence and epidemiological characteristics of dyslipidemia among adults of different genders in Xinjiang, China, providing a basis for promoting ideal lipid management among the Uyghur population.

**Methods.** In this retrospective study, we included 7,646 Uyghur adults from the 2021 physical examination data in Hotan, Xinjiang, and followed up with all participants in 2023 for analysis. Participants completed lifestyle and medical history questionnaires and underwent lipid profiling. Dyslipidemia was defined according to the 2023 Chinese guidelines for lipid management. Group differences were analyzed using $t$-tests, ANOVA, and chi-square tests, the trend test for ordered categorical variables was conducted using univariate linear regression, and multivariable logistic regression was performed to explore risk factors for dyslipidemia.

**Results.** In 2023, the average levels of waist circumference, fasting glucose, total cholesterol (TC), high-density lipoprotein cholesterol (HDL-C), and non-HDL-C among Uyghur men and women, as well as the prevalence of diabetes, increased compared to 2021, with significant statistical differences within the same gender groups ($P < 0.001$). The primary types of dyslipidemia among Uyghur adults were low HDL-C. After age and gender standardization, the overall standardized prevalence of high low-density lipoprotein cholesterol (LDL-C), hypertriglyceridemia, and low HDL-C showed a downward trend, with a more pronounced decrease among men. The prevalence of hypercholesterolemia and high non-HDL-C increased from 2021, with a greater increase among women. In 2023, the standardized prevalence rates of hypercholesterolemia, high LDL-C, low HDL-C, hypertriglyceridemia, and high non-HDL-C were higher in women than in men. Multivariable logistic regression adjusted for multiple factors indicated that higher educational attainment (OR 1.992; 95% CI [1.042–3.808]; $P = 0.037$), overweight (OR 1.303; 95% CI [1.085–1.566]; $P = 0.005$), obesity (OR 1.520; 95% CI [1.226–1.886]; $P = 0.000$), and central obesity (OR 1.013; 95% CI [1.006–1.021]; $P = 0.001$) were associated with dyslipidemia in Uyghur men, while in Uyghur women, dyslipidemia prevalence was mainly related to obesity (OR 1.549; 95% CI [1.261–1.902]; $P = 0.000$) and central obesity (OR 1.009; 95% CI [1.002–1.016]; $P = 0.01$).

**Conclusion.** The primary forms of dyslipidemia among Uyghur adults include low HDL-C levels. The prevalence of high LDL-C, hypertriglyceridemia, and low HDL-C is showing a declining trend, particularly among men. In contrast, the prevalence of

---

hypercholesterolemia and high non-HDL-C is increasing more significantly among women. In men, risk factors for dyslipidemia include higher educational attainment, being overweight, obesity, and central obesity. In women, the prevalence of dyslipidemia is mainly associated with obesity and central obesity.

# INTRODUCTION

Despite new advancements in our understanding of the mechanisms driving atherosclerotic cardiovascular disease (ASCVD) and decades of innovations in medical and interventional cardiovascular care, ASCVD remains the leading cause of death for most ethnic groups of both men and women (*Medzikovic et al., 2023*; *Altschmiedová et al., 2022*; *Banach et al., 2023*; *Kosmas et al., 2022*; *Man, Beckman & Jaffe, 2020*), and dyslipidemia is closely related to the development of atherosclerotic diseases (*Ray et al., 2022*; *Mszar et al., 2024*; *Tabassum et al., 2022*). Dyslipidemia, which includes elevated levels of total cholesterol (TC), low-density lipoprotein cholesterol (LDL-C), and triglycerides (TG), and decreased concentrations of high-density lipoprotein cholesterol (HDL-C) (*Abuzhalihan et al., 2019*), is closely associated with the development of atherosclerotic diseases.In addition, the importance of non-high-density lipoprotein cholesterol (non-HDL-C) has recently garnered significant attention. Non-HDL-C is defined as the total cholesterol content of all lipoproteins excluding high-density lipoprotein (HDL), including LDL-C, triglyceride-rich lipoproteins (TRL), TRL remnants, and lipoprotein(a) (*Lin et al., 2025*). The 2016 ESC/EAS Guidelines for the Management of Dyslipidemias also indicate that non-HDL-C is a better predictor of cardiovascular disease mortality than LDL-C and recommend reducing non-HDL-C as a secondary lipid-lowering target (*Zhang et al., 2025*). Dyslipidemia is a modifiable risk factor that can reduce the incidence of ASCVD by managing lipid abnormalities (*Capece et al., 2024*; *Reamy, Ford & Goodman, 2024*).

Long-standing research has been conducted on the lipid profiles of East Asian (EAS) and Western European populations (*Brunham, Lonn & Mehta, 2024*; *Dybiec et al., 2023*), and numerous confounders and random biases, such as population genetic structure, lifestyle habits, and dietary patterns, have been identified between different ethnicities. Unlike other ethnic groups in China, such as the Han, the Uyghur population of Xinjiang possesses both Western and Eastern Eurasian genetic components (*Dybiec et al., 2023*; *Ning et al., 2023*; *Bian et al., 2016*). High-fat and high-energy dietary habits, along with the consumption of dairy products and red meat, have fostered a unique gut microbiome in the Uyghur population, which in turn affects the occurrence of many related diseases (*Guo et al., 2022*; *Cheng et al., 2021*). Furthermore, there are differences in lipid profiles between males and females (*Tabassum et al., 2022*), and results from the Framingham study also suggest that gender differences in the development or complications of atherosclerosis may be due to the dimorphic nature of major modifiable risk factors, including smoking, dyslipidemia,

hypertension, and diabetes (*Man, Beckman & Jaffe, 2020*). However, understanding and exploration of gender differences in detailed lipid profiles among the Uyghur population of Xinjiang are limited. This study aims to use the 2021 and 2023 national physical examination data from the Hotan area of Xinjiang to estimate the prevalence and trends of dyslipidemia among Uyghur adults of different genders and analyze its associated risk factors, thus providing a reliable basis for adopting distinct measures to promote ideal lipid management among different gender groups of the Uyghur population.

## MATERIALS & METHODS

### Study participants

Uyghur residents aged ≥18 years from Moyu County in Hotan, Xinjiang, who participated in the national physical examination in 2021 were selected as study subjects. The inclusion and exclusion criteria were as follows: Inclusion criteria: (1) complete demographic data such as gender, age, and ethnicity; (2) complete data on blood pressure, height, weight, and waist circumference; (3) age ≥18 years; (4) complete data on lipids, fasting blood glucose, and kidney function. Exclusion criteria: patients with malignant tumors, severe liver or kidney dysfunction, and those unwilling to participate in the study. A total of 7,646 participants were included and followed up in 2023. This study adhered to the Declaration of Helsinki and was approved by the Ethics Committee of the First Affiliated Hospital of Xinjiang Medical University (240424-14 and 220525-06-2305A-YI). All participants signed informed consent forms.

### Data collection

(1) Questionnaires: Professionally trained personnel collected general information, disease history, personal history, and data on control and health services from participants using a standardized questionnaire. (2) Physical examinations: Measurements of height, weight, waist circumference, blood pressure, and heart rate were taken. (3) Laboratory tests: five ml of fasting venous blood was drawn from study participants, and fasting blood glucose (FBG), TC, TG, HDL-C, and LDL-C were measured using an automatic biochemical analyzer (XC 8001, Sichuan New Health Biotechnology Co., Ltd., China). Non-HDL-C = Total Cholesterol (TC) − High-Density Lipoprotein Cholesterol (HDL-C).

### Indicator definitions and assessment standards

According to the 2023 Chinese guidelines for lipid management, LDL-C < 2.6 mmol/L and non-HDL-C < 3.4 mmol/L is considered optimal. TC < 5.2 mmol/L, TG < 1.7 mmol/L, LDL-C < 3.4 mmol/L, and non-HDL-C < 4.1 mmol/L are defined as desirable levels 5.2≤TC < 6.2 mmol/L, 3.4≤LDL-C < 4.1 mmol/L, 1.7≤TG < 2.3 mmol/L, and 4.1≤non-HDL-C < 4.9 mmol/L are considered borderline elevated. Non-HDL-C ≥ 4.9 mmol/L is defined as high non-high density lipoprotein cholesterolemia(high non-HDL-C). Dyslipidemia was defined as TG ≥ 2.3 mmol/L, TC ≥ 6.2 mmol/L, LDL ≥ 4.1 mmol/L, HDL < 1.0 mmol/L or self-reported use of lipid-lowering medications. Body mass index (BMI) < 18.5 kg/m$^2$ is considered underweight, 18.5 kg/m$^2$ ≤ BMI < 24 kg/m$^2$ is normal, 24 kg/m$^2$ ≤ BMI < 28 kg/m$^2$ is overweight, and BMI ≥ 28.0 kg/m$^2$ is obese (*Pan, Wang & Pan, 2021*). A

waist circumference of < 90 cm for men and < 85 cm for women is normal, while ≥ 90 cm for men and ≥ 85 cm for women indicates central obesity (*Opoku et al., 2019*).

## Statistical analysis

Statistical analyses were conducted using SPSS 27.0 and Graph Pad Prism 9.5.0. Data conforming to a normal distribution are presented as mean ± standard deviation and compared using the *t*-test. Data not conforming to a normal distribution are presented as median (Q1, Q3) and compared using the Mann–Whitney *U* test. Categorical data are presented as cases (%) and compared using the chi-square test. The trend test for ordered categorical variables was conducted using univariate linear regression. Multivariable logistic regression was used to analyze potential factors associated with dyslipidemia. The prevalence of dyslipidemia was standardized using data from the Sixth National Population Census of China in 2010. A *p*-value <0.05 was considered statistically significant.

# RESULTS

## Basic characteristics of participants

Among the survey participants, there were fewer males (3,198) compared to females (4,448), with males having a higher average age of $59.91 \pm 10.20$ years compared to females at $57.41 \pm 9.43$ years, showing a statistically significant age difference ($P < 0.001$). Females exhibited higher average levels of BMI, heart rate, fasting blood glucose, TC, TG, HDL-C, LDL-C, and non-HDL-C as well as a higher prevalence of hypertension compared to males, with all *P* values < 0.05 (Table 1). Smoking and alcohol consumption were predominantly observed among males, and the number of males who smoked and drank alcohol decreased in 2023 compared to 2021 ($p < 0.001$) (Tables 1 and 2). In 2023, both males and females in the region showed increases in waist circumference, heart rate, fasting blood glucose, TC, HDL-C and non-HDL-C from 2021, with significant statistical differences within the same gender groups ($P < 0.001$) (Table 2). In 2021, the prevalence of coronary heart disease showed no statistically significant difference between gender groups ($P = 0.374$). However, in 2023, an increase in prevalence was observed in both males and females, with a statistically significant difference in the female group ($P = 0.002$) (Tables 1 and 2).

## Distribution of lipid levels among participants

There was a correlation between indicators such as TC, LDL-C, HDL-C, non-HDL-C and age, with all *P*-values <0.05 (Table 3). In 2021, the peak average values of TC and non-HDL-C were observed in the 60–69 age group, but by 2023, the peak had shifted forward to the 50–59 age group, and the lipids levels in 2023 were higher than those in 2021, with significant differences between male and female groups, $P < 0.01$. In both 2021 and 2023, the peak average values of LDL-C were observed in the 60–69 age group, and the average lipids levels in 2023 were lower than those in 2021, with statistically significant differences between male and female groups, $P < 0.01$ (Tables 2 and 3).The main types of dyslipidemia among Uyghur adults were low HDL-C (Tables 4 and 5). After standardizing for age and gender, the prevalence of hypercholesterolemia increased from 0.8 in 2021 to 3.8 in 2023, with a more notable change in females (from 0.7% in 2021

**Table 1  Baseline data of respondents in 2021.**

| Characteristic | Total (n = 7,646) | Men (n = 3,198) | Women (4,448) | P value |
|---|---|---|---|---|
| Age, mean ± SD, years | 58.45 ± 9.84 | 59.91 ± 10.20 | 57.41 ± 9.43 | <0.001 |
| Waist circumference, cm | 92.06 ± 11.70 | 92.77 ± 11.09 | 91.55 ± 12.09 | <0.001 |
| Body mass index, mean ± SD, kg/m$^2$ | 26.39 ± 4.51 | 25.77 ± 3.93 | 26.84 ± 4.84 | <0.001 |
| Heart rate, mean ± SD, beats/min | 79.17 ± 9.24 | 77.59 ± 9.68 | 80.30 ± 8.74 | <0.001 |
| Blood pressure, mean ± SD, mmHg | | | | |
| Systolic blood pressure | 118.77 ± 14.83 | 117.66 ± 13.27 | 119.57 ± 15.81 | <0.001 |
| Diastolic blood pressure | 76.31 ± 9.90 | 75.31 ± 9.11 | 77.03 ± 10.37 | <0.001 |
| Current smoking | 460 (6.02) | 456 (14.26) | 4 (0.09) | <0.001 |
| Current drinking | 156 (2.04) | 156 (4.88) | 0 | <0.001 |
| Fasting plasma glucose (IQR), mmol/L | 4.68[4.33,5.15] | 4.67[4.31,5.16] | 4.68[4.34,5.15] | 0.248 |
| Serum creatinine, mean ± SD, μmol/L | 54.63 ± 16.35 | 62.69 ± 18.96 | 48.83 ± 10.99 | <0.001 |
| Blood urea nitrogen, mean ± SD, mmol/L | 4.65 ± 1.43 | 4.94 ± 1.45 | 4.45 ± 1.39 | <0.001 |
| Total cholesterol, mean ± SD, mmol/L | 3.68 ± 1.18 | 3.55 ± 1.14 | 3.78 ± 1.2 | 0.108 |
| Triglycerides, median (IQR), mmol/L | 1.21[0.92,1.67] | 1.19[0.89,1.64] | 1.24[0.94,1.70] | <0.001 |
| High-density lipoprotein, mean ± SD, mmol/L | 0.99[0.91,1.10] | 0.96[0.89,1.06] | 1.02[0.93,1.12] | <0.001 |
| Low-density lipoprotein, mean ± SD, mmol/L | 2.67 ± 0.94 | 2.60 ± 0.96 | 2.72 ± 0.92 | 0.038 |
| Non-high-density lipoprotein, mean ± SD, mmol/L | 2.57 ± 0.83 | 2.49 ± 0.87 | 2.63 ± 0.81 | <0.001 |
| Past medical history | | | | |
| Coronary heart disease, n (%) | 295 (3.86) | 116 (3.63) | 179 (4.02) | 0.374 |
| Hypertension, n (%) | 2,345 (30.67) | 877 (27.42) | 1,468 (33.00) | <0.001 |
| Diabetes mellitus, n (%) | 398 (5.21) | 161 (5.03) | 237 (5.33) | 0.568 |

to 6.0% in 2023),and the prevalence of high non-HDL-C increased from 0.2% in 2021 to 2.7% in 2023. The overall standardized prevalence of high LDL-C, low HDL-C, and hypertriglyceridemia showed a declining trend, particularly among males. The increase in hypertriglyceridemia prevalence among females from 2021 to 2023 was minimal, and in 2023, females showed higher standardized prevalence rates of hypercholesterolemia, high LDL-C, low HDL-C, hypertriglyceridemia, and high non-HDL-C compared to males (Table 4). However, when comparing crude prevalence rates of dyslipidemia between genders, males showed higher rates than females, and $P$-values were < 0.001.The crude prevalence rate of hypercholesterolemia, hypertriglyceridemia, and high non-HDL-C showed no significant difference between genders, and all $P$-values were $e > 0.05$ (Table 5, Fig. 1).

## Analysis of related factors

The prevalence of low HDL-C and dyslipidemia was higher among males compared to females ($P < 0.05$). In 2021, the peak prevalence rates of various types of dyslipidemia were predominantly observed in the 50–59 age group. However, by 2023, the peak had shifted forward and was identified in the 40–49 age group (Table 5, Fig. 1). In both 2021 and 2023, there was no statistically significant difference in the crude prevalence rate of high non-HDL-C across age groups ($p > 0.05$), but a peak was observed in the 50–59 age

Maimaitituersun et al. (2025), *PeerJ*, DOI 10.7717/peerj.19344

**Table 2  Basic characteristics of survey subjects after 2 years of follow-up.**

| Characteristic | Men (*n* = 3,198) | | | Women (4,448) | | |
|---|---|---|---|---|---|---|
| | 2021 | 2023 | *P* value | 2021 | 2023 | *P* value |
| Waist circumference, cm | 92.77 ± 11.09 | 93.89 ± 11.06 | <0.001 | 91.55 ± 12.09 | 91.77 ± 11.94 | 0.396 |
| Body mass index, mean ± SD, kg/m² | 25.77 ± 3.93 | 25.69 ± 3.93 | 0.407 | 26.84 ± 4.84 | 26.71 ± 4.88 | 0.202 |
| Heart rate, mean ± SD, beats/min | 77.59 ± 9.68 | 80.93 ± 7.48 | <0.001 | 80.30 ± 8.74 | 82.57 ± 7.21 | <0.001 |
| Blood pressure, mean ± SD, mmHg | | | | | | |
| Systolic blood pressure | 117.66 ± 13.27 | 120.29 ± 10.97 | <0.001 | 119.57 ± 15.81 | 118.60 ± 11.61 | 0.001 |
| Current smoking | 456 (14.26) | 210 (6.57) | <0.001 | 4 (0.09) | 0 | 0.125 |
| Current drinking | 156 (4.88) | 36 (1.13) | <0.001 | 0 | 0 | – |
| Diastolic blood pressure | 75.31 ± 9.11 | 72.87 ± 6.80 | <0.001 | 77.03 ± 10.37 | 75.72 ± 8.61 | <0.001 |
| Fasting plasma glucose (IQR), mmol/L | 4.67[4.31,5.16] | 4.80[4.40,5.40] | <0.001 | 4.68[4.34,5.15] | 4.85[4.45,5.50] | <0.001 |
| Serum creatinine, mean ± SD, μmol/L | 62.69 ± 18.96 | 59.63 ± 16.81 | <0.001 | 48.83 ± 10.99 | 46.11 ± 12.35 | <0.001 |
| Blood urea nitrogen, mean ± SD, mmol/L | 4.94 ± 1.45 | 5.17 ± 1.40 | <0.001 | 4.45 ± 1.39 | 4.87 ± 1.35 | <0.001 |
| Total cholesterol, mean ± SD, mmol/L | 3.55 ± 1.14 | 4.03 ± 1.03 | <0.001 | 3.78 ± 1.2 | 4.21 ± 1.03 | <0.001 |
| Triglycerides, median (IQR), mmol/L | 1.19[0.89,1.64] | 1.04[0.74,1.37] | <0.001 | 1.24[0.94,1.70] | 1.07[0.77,1.40] | <0.001 |
| High-density lipoprotein, mean ± SD, mmol/L | 0.96[0.89,1.06] | 1.11[0.97,1.30] | <0.001 | 1.02[0.93,1.12] | 1.19[1.02,1.37] | <0.001 |
| Low-density lipoprotein, mean ± SD, mmol/L | 2.60 ± 0.96 | 2.10 ± 0.79 | <0.001 | 2.72 ± 0.92 | 2.10 ± 0.79 | <0.001 |
| Non-high-density lipoprotein, mean ± SD, mmol/L | 2.49 ± 0.87 | 2.86 ± 0.96 | <0.001 | 2.63 ± 0.81 | 2.99 ± 0.93 | <0.001 |
| Past medical history | | | | | | |
| Coronary heart disease, *n* (%) | 116 (3.63) | 144 (4.50) | 0.076 | 179 (4.02) | 242 (5.44) | 0.002 |
| Hypertension, *n* (%) | 877 (27.42) | 917 (28.67) | 0.266 | 1,468 (33.00) | 1,513 (34.02) | 0.312 |
| Diabetes mellitus, *n* (%) | 161 (5.03) | 257 (8.04) | <0.001 | 237 (5.33) | 323 (7.26) | <0.001 |

Maimaitituersun et al. (2025), *PeerJ*, DOI 10.7717/peerj.19344

**Table 3  Blood lipid distribution between different age groups.**

| Age | 2021 | | | | | 2023 | | | | |
|---|---|---|---|---|---|---|---|---|---|---|
| | TC | TG | LDL-C | HDL-C | non-HDL-C | TC | TG | LDL-C | HDL-C | non-HDL-C |
| 18–29 | 3.09 ± 0.72 | 0.68[0.39,0.88] | 2.17 ± 0.74 | 0.95[0.89,1.05] | 2.06 ± 0.70 | 3.82 ± 0.82 | 0.69[0.29,1.01] | 1.93 ± 0.72 | 1.08[0.94,1.25] | 2.74 ± 0.85 |
| 30–39 | 2.92 ± 0.81 | 1.23[0.81,1.49] | 2.02 ± 0.80 | 1.03[0.88,1.20] | 1.88 ± 0.97 | 4.58 ± 1.20 | 1.08[0.84,1.52] | 2.35 ± 0.83 | 0.99[0.93,1.37] | 3.45 ± 1.00 |
| 40–49 | 3.62 ± 0.89 | 1.25[0.91,1.74] | 2.66 ± 0.94 | 1.01[0.92,1.17] | 2.41 ± 0.80 | 4.13 ± 0.90 | 1.00[0.72,1.32] | 2.07 ± 0.78 | 1.13[0.99,1.33] | 2.94 ± 0.90 |
| 50–59 | 3.68 ± 0.97 | 1.23[0.93,1.73] | 2.65 ± 0.94 | 0.99[0.91,1.09] | 2.58 ± 0.84 | 4.18 ± 0.93 | 1.03[0.73,1.39] | 2.11 ± 0.77 | 1.14[0.99,1.34] | 2.99 ± 0.92 |
| 60–69 | 3.75 ± 0.96 | 1.20[0.93,1.63] | 2.71 ± 0.94 | 1.00[0.92,1.10] | 2.65 ± 0.83 | 4.16 ± 0.97 | 1.07[0.78,1.40] | 2.14 ± 0.81 | 1.16[0.99,1.36] | 2.95 ± 0.95 |
| ≥70 | 3.71 ± 0.93 | 1.20[0.92,1.55] | 2.71 ± 0.93 | 0.98[0.91,1.10] | 2.64 ± 0.85 | 3.99 ± 1.06 | 1.13[0.83,1.42] | 2.04 ± 0.80 | 1.17[1.02,1.36] | 2.77 ± 1.04 |
| P value | <0.001 | 0.007 | <0.001 | <0.001 | <0.001 | <0.001 | <0.001 | 0.284 | 0.002 | <0.001 |

**Notes.**

$P$-value represents the trend test for blood lipid levels among different age groups.

**Table 4 Distribution of blood lipid levels of the respondents (%).**

| Classification | | 2021 | | | 2023 | | |
| --- | --- | --- | --- | --- | --- | --- | --- |
| | | Overall | Men | Women | Overall | Men | Women |
| TC | Normal | 96.7 | 97.7 | 97.4 | 83.6 | 81.3 | 86.0 |
| | Marginally elevated | 2.5 | 1.7 | 1.9 | 12.5 | 17.0 | 7.9 |
| | hypercholesterolemia | 0.8 | 0.5 | 0.7 | 3.8 | 1.7 | 6.0 |
| TG | Normal | 82.96 | 81.8 | 84.1 | 93.2 | 95.5 | 90.8 |
| | Marginally elevated | 9.62 | 9.6 | 9.6 | 2.7 | 2.5 | 2.9 |
| | Hypertriglyceridemia | 7.4 | 8.6 | 6.2 | 4.1 | 2 | 6.3 |
| HDL-C | Lower | 55.64 | 66.52 | 44.48 | 33.0 | 30.8 | 35.1 |
| | Normal | 44.36 | 33.48 | 55.52 | 67.0 | 69.2 | 64.9 |
| LDL-C | Ideal level | 59.9 | 59.7 | 60.2 | 73.6 | 80.1 | 66.9 |
| | Normal | 89.5 | 89.9 | 89.1 | 95.3 | 95.9 | 94.6 |
| | Marginally elevated | 8.0 | 7.8 | 8.3 | 3.8 | 3.6 | 4.0 |
| | Hyper-LDL cholesterolemia | 2.5 | 2.3 | 2.7 | 1.0 | 0.5 | 1.4 |
| Non HDL-C | Ideal level | 93 | 93.4 | 92.7 | 69.1 | 64.2 | 74.1 |
| | Normal | 98.6 | 98.6 | 98.6 | 85.8 | 83.8 | 87.8 |
| | Marginally elevated | 1.2 | 1.2 | 1.2 | 11.5 | 14.3 | 8.7 |
| | High non-HDL-C | 0.2 | 0.2 | 0.2 | 2.7 | 1.9 | 3.5 |

**Notes.**

*Normal: TC<5.2 mmol/L, LDL-C<3.4 mmol/L, TG<1.7 mmol/L, and non HDL-C<4.1 mmol/L.

Ideal, LDL-C <2.6 mmol/L, and non HDL-C<3.4 mmol/L.

Marginally elevated $5.2 \leq$ TC <6.2 mmol/L, $3.4 \leq$ LDL-C <4.1 mmol/L, $1.7 \leq$ TG <2.3 mmol/L, and $4.1 \leq$ non HDL-C<4.9 mmol/L.

TC $\geq$6.2 mmol/L is considered hypercholesterolemia, TG $\geq$2.3 mmol/L is hypertriglyceridemia, HDL-C <1.0 mmol/L is low HDL-C, LDL-C $\geq$4.1 mmol/L is Hyper-LDL cholesterolemia, and non HDL-C$\geq$4.9 mmol/L is High non-HDL-C.

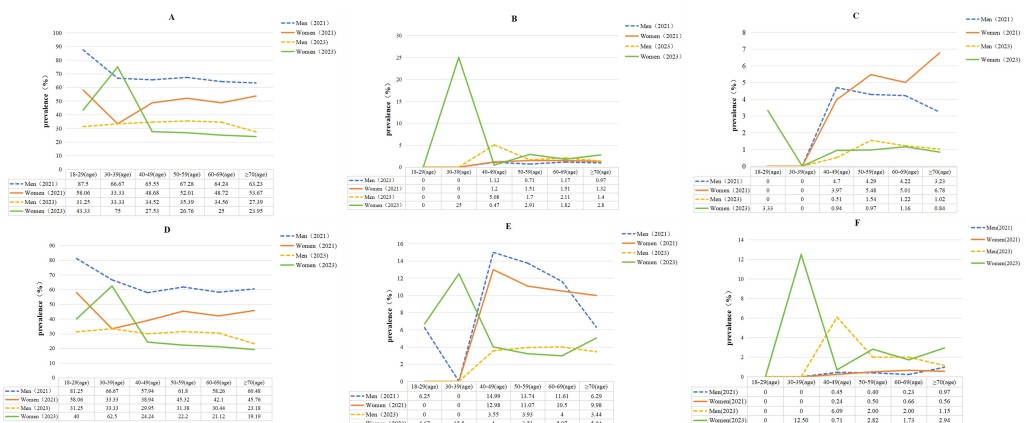

**Figure 1 Trends in the prevalence of dyslipidemia (%).** (A) The crude prevalence rate of dyslipidemia. (B) The crude prevalence rate of Hypercholesterolemia. (C) The crude prevalence rate of high LDL-C. (D) The crude prevalence rate of low HDL-C. (E) The crude prevalence rate of hypertriglyceridemia. (F) The crude prevalence rate of high non-HDL-C.

group (Table 5, Fig. 1). The highest rates of dyslipidemia and low HDL-C were observed in unmarried individuals and smokers, with significant differences between the groups ($P < 0.05$). After adjusting for factors such as gender, age, education, BMI, and habits

Maimaitituersun et al. (2025), *PeerJ*, DOI 10.7717/peerj.19344

**Table 5** The prevalence of dyslipidemia in the population with different characteristics of the respondents.

| Variable | Total (number) | High TC (number (%)) | High LDL-C (number (%)) | Low HDL-C (number (%)) | High TG (number (%)) | Dyslipidemia (number (%)) | High non-HDL-C (number (%)) |
|---|---|---|---|---|---|---|---|
| | | | | **2021** | | | |
| Gender | | | | | | | |
| Male | 3,198 | 30 (0.9) | 131 (4.1) | 1,924 (60.2) | 379 (11.9) | 2,172 (67.9) | 15 (0.5) |
| Female | 4,448 | 63 (1.4) | 231 (5.2) | 1,934 (43.5) | 492 (11.1) | 2,395 (53.8) | 22 (0.5) |
| P value | | 0.06 | 0.026 | <0.001 | 0.283 | <0.001 | 0.874 |
| Age (years) | | | | | | | |
| 18 ∼ 29 | 47 | 0 | 0 | 31 (66.0) | 1 (2.1) | 33 (70.2) | 0 |
| 30 ∼ 39 | 12 | 0 | 0 | 5 (41.7) | 0 | 6 (50.0) | 0 |
| 40 ∼ 49 | 1,279 | 15 (1.2) | 54 (4.2) | 583 (45.6) | 175 (13.7) | 721 (56.4) | 4 (0.3) |
| 50 ∼59 | 3,247 | 39 (1.2) | 163 (5.0) | 1,679 (51.7) | 393 (12.1) | 1,982 (61.0) | 15 (0.5) |
| 60 ∼69 | 1,910 | 26 (1.4) | 89 (4.7) | 942 (49.3) | 210 (11.0) | 1,112 (58.2) | 9 (0.5) |
| ≥70 | 1,151 | 13 (1.1) | 56 (4.9) | 618 (53.7) | 92 (8.0) | 713 (61.9) | 9 (0.8) |
| P value | | 0.949 | 0.503 | <0.001 | <0.001 | 0.010 | 0.665 |
| Education | | | | | | | |
| Elementary school and below | 6,328 | 73 (1.2) | 291 (4.6) | 3,166 (50.0) | 692 (10.9) | 3,746 (59.2) | 0 |
| Junior or senior high school | 1,239 | 19 (1.5) | 63 (5.1) | 651 (52.5) | 165 (13.3) | 768 (62.0) | 32 (2.6) |
| College or above | 79 | 1 (1.0) | 8 (10.1) | 41 (51.9) | 14 (17.7) | 53 (67.1) | 5 (6.3) |
| P value | | 0.536 | 0.058 | 0.262 | 0.011 | 0.076 | <0.001 |
| Marital status | | | | | | | |
| Never married | 42 | 0 | 0 | 30 (71.4) | 1 (2.4) | 32 (76.2) | 0 |
| Married | 6,989 | 82 (1.2) | 320 (4.6) | 3,557 (50.9) | 807 (11.5) | 4,199 (60.1) | 32 (0.5) |
| Divorced or Widowed | 615 | 11 (1.8) | 42 (6.8) | 271 (44.1) | 63 (10.2) | 336 (54.6) | 5 (0.8) |
| P value | | 0.316 | 0.015 | <0.001 | 0.114 | 0.003 | 0.430 |
| BMI | | | | | | | |
| <18.5 | 101 | 0 | 5 (5.0) | 40 (39.6) | 2 (2.0) | 44 (43.6) | 0 |
| 18.5–23.9 | 2,283 | 26 (1.1) | 79 (3.5) | 950 (41.6) | 143 (6.3) | 1,149 (50.3) | 13 (0.6) |
| 24.0–27.9 | 2,617 | 29 (1.1) | 130 (5.0) | 1,363 (52.1) | 282 (10.8) | 1,588 (60.7) | 7 (0.3) |
| ≥28 | 2,645 | 38 (1.4) | 148 (5.6) | 1,505 (56.9) | 444 (16.8) | 1,786 (67.5) | 17 (0.6) |
| P value | | 0.443 | 0.005 | <0.001 | <0.001 | <0.001 | 0.190 |
| Waist | | | | | | | |
| Men<90 or Women<85 | 2,529 | 33 (1.3) | 88 (3.5) | 1,057 (41.8) | 168 (6.6) | 1,276 (50.5) | 12 (0.5) |
| Men≥90 or Women≥85 | 5,117 | 60 (1.2) | 274 (5.4) | 2,801 (54.7) | 703 (13.7) | 3,291 (64.3) | 25 (0.5) |
| P value | | 0.62 | <0.001 | <0.001 | <0.001 | <0.001 | 0.934 |

**Table 5** (*continued*)

| Variable | Total (number) | High TC (number (%)) | High LDL-C (number (%)) | Low HDL-C (number (%)) | High TG (number (%)) | Dyslipidemia (number (%)) | High non-HDL-C (number (%)) |
|---|---|---|---|---|---|---|---|
| Smoking status | | | | | | | |
| Smoking | 460 | 5 (1.1) | 25 (5.4) | 279 (60.7) | 65 (14.1) | 311 (67.6) | 2 (0.4) |
| Non-smoking | 7,186 | 88 (1.2) | 337 (4.7) | 3,579 (49.8) | 806 (11.2) | 4,256 (59.2) | 35 (0.5) |
| *P* value | | 0.794 | 0.466 | <0.001 | 0.056 | <0.001 | 1.000 |
| Drinking status, *n* (%) | | | | | | | |
| Drinking | 156 | 2 (1.3) | 12 (7.7) | 93 (59.6) | 28 (17.9) | 106 (67.9) | 1 (0.6) |
| Non-drinking | 7,490 | 91 (1.2) | 350 (4.7) | 3,765 (50.3) | 843 (11.3) | 4,461 (59.6) | 36 (0.5) |
| *P* value | | 0.715 | 0.079 | 0.021 | 0.009 | 0.034 | 0.534 |
| **2023** | | | | | | | |
| Gender | | | | | | | |
| Male | 3,198 | 62 (1.9) | 40 (1.3) | 928 (29.0) | 121 (3.8) | 1,059 (33.1) | 65 (2.0) |
| Female | 4,448 | 106 (2.4) | 45 (1.0) | 970 (21.8) | 158 (3.6) | 1,161 (26.1) | 104 (2.3) |
| *P* value | | 0.191 | 0.325 | <0.001 | 0.594 | <0.001 | 0.370 |
| Age (years) | | | | | | | |
| 18 ~ 29 | 46 | 0 | 1 (2.2) | 17 (37.0) | 2 (4.3) | 18 (39.1) | 0 |
| 30 ~ 39 | 11 | 2 (18.2) | 0 | 6 (54.5) | 1 (9.1) | 7 (63.6) | 1 (9.1) |
| 40 ~ 49 | 622 | 12 (1.9) | 14 (2.3) | 162 (26.0) | 24 (3.9) | 185 (29.7) | 15 (2.4) |
| 50 ~ 59 | 3,356 | 82 (2.4) | 36 (1.1) | 864 (25.7) | 117 (3.5) | 1,010 (30.1) | 84 (2.5) |
| 60 ~ 69 | 2,112 | 41 (1.9) | 24 (1.1) | 530 (25.1) | 72 (3.4) | 614 (29.1) | 39 (1.8) |
| ≥70 | 1,499 | 31 (2.1) | 10 (0.7) | 319 (21.3) | 63 (4.2) | 386 (25.8) | 30 (2.0) |
| *P* value | | 0.007 | 0.058 | <0.001 | 0.704 | 0.002 | 0.261 |

Maimaitituersun et al. (2025), *PeerJ*, DOI 10.7717/peerj.19344

**Table 5** (*continued*)

| Variable | Total (number) | High TC (number (%)) | High LDL-C (number (%)) | Low HDL-C (number (%)) | High TG (number (%)) | Dyslipidemia (number (%)) | High non-HDL-C (number (%)) |
|---|---|---|---|---|---|---|---|
| Education | | | | | | | |
| Elementary school and below | 6,328 | 135 (2.1) | 69 (1.1) | 1,533(24.2) | 218 (3.4) | 1,790 (28.3) | 137 (2.2) |
| Junior or senior high school | 1,239 | 29 (2.3) | 15 (1.2) | 336 (27.1) | 54 (4.4) | 394 (31.8) | 29 (2.3) |
| College or above | 79 | 4 (5.1) | 1 (1.3) | 29 (36.7) | 7 (8.9) | 36 (45.6) | 3 (3.8) |
| *P* value | | 0.196 | 0.926 | 0.005 | 0.013 | <0.001 | 0.583 |
| Marital status | | | | | | | |
| Never married | 42 | 0 | 1 (2.4) | 15 (35.7) | 1 (2.4) | 16 (38.1) | 0 |
| Married | 6,989 | 150 (2.1) | 76 (1.1) | 1,752 (25.1) | 253 (3.6) | 2,048 (29.3) | 152 (2.2) |
| Divorced or Widowed | 615 | 18 (2.9) | 8 (1.3) | 131 (21.3) | 25 (4.1) | 156 (25.4) | 17 (2.8) |
| *P* value | | 0.279 | 0.653 | 0.030 | 0.774 | 0.051 | 0.394 |
| BMI | | | | | | | |
| <18.5 | 154 | 3 (1.9) | 5 (3.2) | 23 (14.9) | 2 (2.0) | 27 (17.5) | 3 (3.0) |
| 18.5–23.9 | 2,377 | 48 (2.0) | 58 (2.4) | 488 (20.5) | 54 (2.4) | 567 (23.9) | 41 (1.8) |
| 24.0–27.9 | 2,671 | 52 (1.9) | 95 (3.6) | 670 (25.1) | 92 (3.5) | 777 (29.1) | 51 (1.9) |
| ≥28 | 2,444 | 65 (2.7) | 121 (5.0) | 717 (29.3) | 131 (5.0) | 849 (34.7) | 74 (2.8) |
| *P* value | | 0.307 | <0.001 | <0.001 | <0.001 | <0.001 | 0.068 |
| Waist | | | | | | | |
| Men<90 or Women<85 | 2,279 | 37 (1.6) | 21 (0.9) | 489 (21.5) | 79 (3.5) | 576 (25.3) | 34 (1.1) |
| Men≥90 or Women≥85 | 5,367 | 131 (2.45) | 64 (1.2) | 1,409 (26.3) | 200 (3.7) | 1,644 (30.6) | 135 (3.0) |
| *P* value | | 0.026 | 0.301 | <0.001 | 0.579 | <0.001 | <0.001 |
| Smoking status | | | | | | | |
| Smoking | 210 | 7 (3.3) | 1 (0.5) | 68 (32.4) | 4 (1.9) | 78 (37.1) | 9 (4.3) |
| Non-smoking | 7,436 | 161 (2.2) | 84 (1.1) | 1,830 (24.6) | 275 (3.7) | 2,142 (28.8) | 160 (2.2) |
| *P* value | | 0.255 | 0.732 | 0.010 | 0.172 | 0.009 | 0.051 |

Maimaitituersun et al. (2025), *PeerJ*, DOI 10.7717/peerj.19344

**Table 5** (*continued*)

| Variable | Total (number) | High TC (number (%)) | High LDL-C (number (%)) | Low HDL-C (number (%)) | High TG (number (%)) | Dyslipidemia (number (%)) | High non-HDL-C (number (%)) |
|---|---|---|---|---|---|---|---|
| Drinking status, *n* (%) | | | | | | | |
| Drinking | 36 | 1 | 0 (0) | 14 (38.9) | 1 (2.8) | 16 (44.4) | 2 (5.6) |
| Non-drinking | 7,610 | 167 | 85 (1.1) | 1,884 (24.8) | 278 (3.7) | 2,204 (29.0) | 167 (2.2) |
| *P* value | | 0.551 | 1.000 | 0.050 | 1.000 | 0.041 | 0.189 |

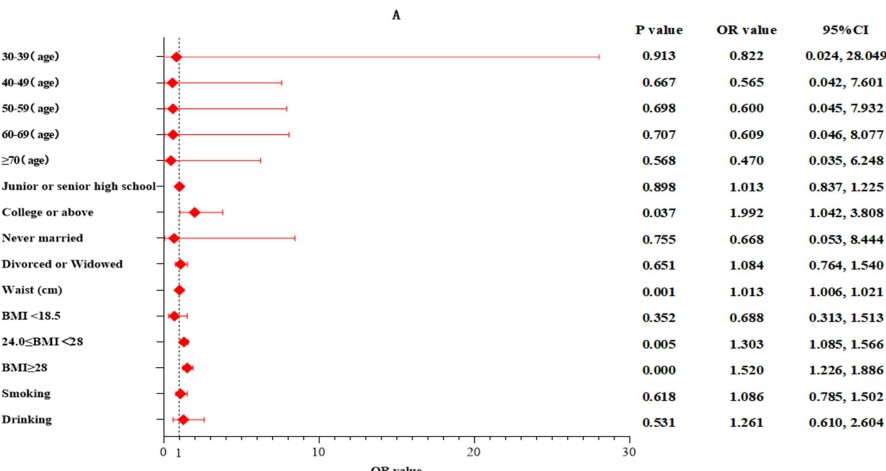

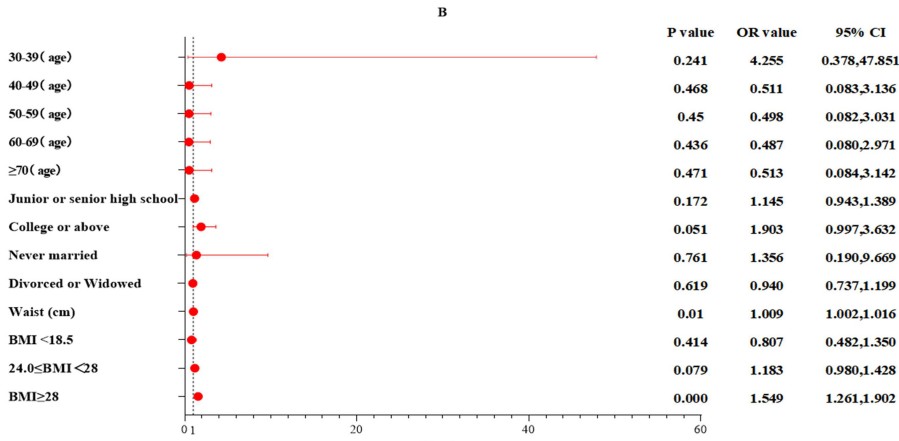

**Figure 2 Results of multivariate logistic regression analysis.** (A) For males, (B) for females. Age reference 18–29 years old; education background reference primary school or below; marital status refer to married; BMI reference normal weight group (18.5 ≤ BMI 28).

of smoking and drinking through multivariable logistic regression, the results indicated that higher educational attainment(contain the college or above level of education) (OR 1.992; 95% CI [1.042–3.808]; $P = 0.037$), overweight (OR 1.303; 95% CI [1.085–1.566]; $P = 0.005$), obesity (OR 1.520; 95% CI [1.226–1.886]; $P = 0.000$), and central obesity (OR 1.013; 95% CI [1.006–1.021]; $P = 0.001$) were associated with dyslipidemia in Uyghur males, while obesity (OR 1.549; 95% CI [1.261–1.902]; $P = 0.000$) and central obesity (OR 1.009; 95% CI [1.002–1.016]; $P = 0.01$) were related to dyslipidemia prevalence in Uyghur females (Fig. 2).

## DISCUSSION

Our findings indicate that dyslipidemia among Uyghur adults is primarily characterized by low high-density lipoprotein cholesterol (HDL-C), with decreasing prevalence rates for high low-density lipoprotein cholesterol (LDL-C), hypertriglyceridemia, and low HDL-C, particularly among males. The standardized prevalence rates of high TC and high non-HDL-C increased compared to 2021 (the prevalence of high TC rose from 0.8 to 3.8, and the prevalence of high non-HDL-C increased from 0.2 to 2.7), with a more pronounced change in prevalence observed among females. Risk factors for dyslipidemia in males include higher educational attainment, being overweight, obesity, and central obesity, whereas in females, dyslipidemia is mainly associated with obesity and central obesity. Long-term studies of lipid profiles in East Asian (EAS) and Western European populations exist (*Brunham, Lonn & Mehta, 2024*; *Dybiec et al., 2023*), yet there is a lack of research on the distribution and influencing factors of lipid levels among different genders of the Uyghur population in Xinjiang. Our results can serve as part of the epidemiological baseline data for dyslipidemia among the Uyghur population, providing a basis for promoting ideal lipid management among different gender groups.

Xinjiang Uygur Autonomous Region is a vast, multi-ethnic area in the northwest of China, predominantly inhabited by the Han, Uygur, and Kazakh ethnic groups. Among these, the Uygur people constitute the largest minority population in Xinjiang, with the majority residing in rural areas. Previous studies on the Uyghur population in Xinjiang reported lower awareness, treatment, and control rates of dyslipidemia compared to other regions, hence a higher prevalence rate than the national average (*Ma et al., 2015*). Our study indicates a decreasing trend in the prevalence of dyslipidemia among adults in the region from 2021 to 2023, likely benefiting from the "Healthy China 2030" initiative (*Bei, Yang & Xiao, 2018*), which has raised awareness about dyslipidemia. Improved medical resource distribution, lifestyle and dietary habits, increased physical activity, and the use of lipid-lowering drugs may have contributed to this control (*Lu et al., 2021*). China's comprehensive healthcare system also plays a protective role in the health of its rural population.

A survey (*Ge et al., 2015*) on the prevalence of dyslipidemia at the grassroots level in China found that dyslipidemia is mainly composed of low HDL-C and hypertriglyceridemia levels, a finding also confirmed by our study in the Uyghur population. We observed higher rates of low HDL-C and dyslipidemia among Uyghur males compared to females, with the highest prevalence among those aged 50–59. The research findings published by *Guo et al. (2014)* indicate that the prevalence rates of hypercholesterolemia, hypertriglyceridemia, high LDL-C, and low HDL-C among the Kazakh ethnic group in Xinjiang, after standardization for gender and age, were 6.9%, 9.3%, 2.9%, and 20.8% respectively. When compared with our study results, it was found that the prevalence rates of hypercholesterolemia, hypertriglyceridemia, and high LDL-C were higher in the Kazakh ethnic group than in the Uygur ethnic group (with standardized prevalence rates of 3.8%, 4.1%, and 1.0% respectively). In an epidemiological survey conducted by *Zhou et al. (2016)*, the prevalence of low HDL-C among the Uygur ethnic group was 24.9%,

which is higher than that of the Kazakh and Han ethnic groups included in the same study (18.3% and 18.2% respectively). In our study results, the standardized prevalence of low HDL-C in the Uygur population in 2023 was 33.0%, which also aligns with the findings of other previously reported studies. The significant differences in HDL-C levels proposed by Rutherford and others may be related to economic development, urbanization, changes in dietary patterns, BMI, and central obesity (*Rutherford et al., 2010*; *Li et al., 2022*). The Uyghur population of Xinjiang, with its blend of Western and Eastern Eurasian ancestry (*Dybiec et al., 2023*; *Ning et al., 2023*), faces increased disease risk due to long-term genetic mixing. The diet in Xinjiang (*Ning et al., 2023*), predominantly red meat and dairy (*Ning et al., 2023*; *Fan et al., 2020*), correlates positively with dyslipidemia risk (*Zhang et al., 2019*). Moreover, higher residential greenery is significantly associated with lower TG levels and higher HDL-C levels (*Fan et al., 2020*). The semi-arid continental climate of Xinjiang, with minimal rainfall and desert-covered lands, contributes to the hypertriglyceridemia and low HDL-C levels in the population. Regarding the differences in blood lipid profiles between the Uyghur and Kazakh ethnic groups in Xinjiang, we believe that lifestyle, dietary habits, and genetic influences play a dominant role. Due to Xinjiang's geographical location, living environment, and ethnic composition being different from other regions in China, the lifestyles of its people also vary significantly. Since most Kazakhs lead a nomadic lifestyle, their consumption of beef, lamb, and dairy products is much higher than that of other ethnic groups. The Uyghur, primarily engaged in agriculture, have a higher consumption level of flour-based foods and meat compared to other ethnic groups. Consequently, there are differences between the two ethnic groups in terms of blood lipid levels, prevalence of hypertension, and cardiovascular diseases, which have also been mentioned in previous studies (*Abuzhalihan et al., 2019*; *Ning et al., 2023*; *Bian et al., 2016*). Gender differences in lipid profiles (*Man, Beckman & Jaffe, 2020*), where males are more than twice as women likely to have low HDL-C across all age groups. Thus, genetic factors, gender, dietary structure, geographic environment, smoking, and drinking history all influence the prevalence of dyslipidemia among the Uyghur population, previous literature (*Fan et al., 2020*) further validate our findings. The exact etiology of dyslipidemia is not clear, and more in-depth studies are needed to explore the specific causes.

It is worth further consideration that our research findings indicate a higher level of education is a relevant risk factor for dyslipidemia among Uyghur men. *Lara & Amigo (2018)* have studied the correlation between educational level and blood lipids, showing that men with lower education levels tend to have better blood lipid levels compared to their highly educated peers. Other research (*Espírito Santo et al., 2022*) findings also suggest a significant positive correlation between socioeconomic status and dyslipidemia, with men and women from higher socioeconomic strata experiencing increased total cholesterol and LDL-c levels even after controlling for confounding factors. Moreover, a higher socioeconomic status is associated with an increased prevalence of hypercholesterolemia, regardless of gender. These conclusions are consistent with our research findings. *Emmel et al. (2021)* propose that the net effect of education is reflected in the ability to translate health-related information into behavior and to enhance the understanding of treatment measures. This may support the hypothesis that in highly educated populations, the genetic
impact on blood lipid indicators such as LDL-C may be stronger due to the ability to create environments with fewer health risks. This is clearly inconsistent with our conclusion, and a possible explanation is that social progress facilitates access to some attractive unhealthy behaviors (fast food diet, physical inactivity, alcohol and tobacco use) (*Espírito Santo et al., 2022*). In our study, unhealthy lifestyles such as smoking and drinking are mainly observed in the male population, which may also be one of the reasons for this opposite conclusion. Secondly, the findings of Macarena and others also suggest that highly educated men have lower physical activity and higher total skinfold thickness, possibly because their occupations require less physical labor, which may increase body fat and potentially lead to elevated blood lipid levels. It is necessary to increase the sample size or use more heterogeneous samples, as the socioeconomic homogeneity of the sample may obscure the relevant impact of education on blood lipid levels, which could be more significant in more heterogeneous conditions (*Lara & Amigo, 2018*).

Dyslipidemia has become a major risk factor for cardiovascular disease (CVD) in China (*Lu et al., 2021*). Studies (*Yang et al., 2021*) using large-scale genetic data suggest a causal relationship between HDL-C $\leq$ 50 mg/dL and CVD risk. Epidemiological research (*Chen et al., 2024*) also highlights HDL-C and TG as strong predictors of CVD, with studies (*Georgoulis et al., 2022*) suggesting TG as a predictor of CVD risk among males, overweight/obese individuals, and those on lipid-lowering medication. While past discussions (*Crea, 2022*; *Ray et al., 2019*; *Ballantyne et al., 2023*) on lipid management have focused on LDL-C treatment, potentially overlooking high-risk ASCVD individuals (*Björnson et al., 2023*), dyslipidemia remains a potential target for reducing CVD incidence. Effective prevention strategies, especially for HDL-C and TG levels, are needed among the Uyghur population in Xinjiang to shift focus towards early prevention strategies, maintaining health, and reducing ASCVD incidence and improving patient outcomes. Since the proposal of the "Healthy China 2030" initiative, the rural medical environment has improved compared to the past. After local medical institutions began providing free oral lipid-lowering medications to patients with hyperlipidemia, the lipid levels of the Uyghur population significantly decreased. However, total cholesterol (TC) and non-high-density lipoprotein cholesterol (non-HDL-C) levels continue to show an upward trend, and the prevalence of low high-density lipoprotein cholesterol (HDL-C) remains high. We believe that it is necessary to strengthen health education on dyslipidemia management among the Uyghur population in southern Xinjiang to enhance their awareness of dyslipidemia. Increasing physical exercise and reducing body weight can also help lower lipid levels to some extent. The high-calorie dietary habits of the Uyghur population are another contributing factor to dyslipidemia, so efforts should be made to promote education and improve unhealthy eating habits. Additionally, for patients who respond poorly to statin therapy or are intolerant to statins, the use of proprotein convertase subtilisin/kexin type 9 (PCSK9) inhibitors and other methods should be actively promoted to improve the treatment rate of dyslipidemia and reduce its prevalence.Additionally, given the higher average levels of BMI, fasting blood glucose, TC, TG, HDL-C, LDL-C, non-HDL-C and the higher prevalence of hypertension among Uyghur females, their metabolic health should be further prioritized and managed.

This study has both strengths and limitations. Firstly, our research focuses on the Uyghur population, which helps to enrich the global understanding of dyslipidemia across different genetic and cultural backgrounds. A relatively large sample size enhances the reliability of the findings, and the two-year longitudinal data provides insights into the dynamics of lipid profile changes. However, there are several limitations. The inclusion of patients was affected by factors such as working or studying away from home, resulting in a smaller number of participants aged 18–39, which limits the representativeness of this age group and introduces some statistical biases. To address this, we conducted a more detailed age-stratified analysis to better describe the lipid levels in each age group. Additionally, all participants in this study were from rural areas, which means the findings may not accurately estimate the lipid levels of urban populations. To address this limitation, we plan to include urban populations in future follow-up surveys, further increase the sample size, and use more heterogeneous samples to improve the representativeness of the study.

## CONCLUSION

Dyslipidemia in Uyghur adults is primarily characterized by low high-density lipoprotein cholesterol (HDL-C), with decreasing prevalence rates of high low-density lipoprotein cholesterol (LDL-C), hypertriglyceridemia, and low HDL-C, particularly more pronounced among males. The increase in hypercholesterolemia and high non-HDL-C rates is greater among females than males. Higher educational attainment, overweight, obesity, and central obesity are associated risk factors for dyslipidemia in males, while in females, dyslipidemia is mainly related to obesity and central obesity. Additionally, Uyghur females exhibit higher average levels of BMI, fasting blood glucose, TC, TG, HDL-C, LDL-C, non-HDL-C and a higher prevalence of hypertension compared to males. Therefore, the metabolic health of Uyghur women should receive further attention and management.

### Funding

This work was funded by the National Key Research and Development Program of China (2021YFC2500605); "Tianshan Talents" training program (2022TSYCL0030). The funders had no role in study design, data collection and analysis, decision to publish, or preparation of the manuscript.

### Grant Disclosures

The following grant information was disclosed by the authors:
National Key Research and Development Program of China: 2021YFC2500605.
"Tianshan Talents" Training Program: 2022TSYCL0030.

### Competing Interests

The authors declare there are no competing interests.

## Author Contributions

- Gulinigaer Maimaitituersun conceived and designed the experiments, performed the experiments, analyzed the data, prepared figures and/or tables, and approved the final draft.
- Subinuer Jureti performed the experiments, prepared figures and/or tables, and approved the final draft.
- Ziyu Yi conceived and designed the experiments, prepared figures and/or tables, and approved the final draft.
- Yaqi Zhou performed the experiments, prepared figures and/or tables, and approved the final draft.
- Meng Li analyzed the data, prepared figures and/or tables, and approved the final draft.
- Mengwei Wei analyzed the data, prepared figures and/or tables, and approved the final draft.
- Ziyang Liu conceived and designed the experiments, authored or reviewed drafts of the article, and approved the final draft.
- Menglong Jin conceived and designed the experiments, analyzed the data, authored or reviewed drafts of the article, and approved the final draft.
- Zhenyan Fu conceived and designed the experiments, authored or reviewed drafts of the article, and approved the final draft.

## Human Ethics

The following information was supplied relating to ethical approvals (i.e., approving body and any reference numbers):

This study complied with the Declaration of Helsinki and was approved by the Ethics Committee of the First Affiliated Hospital of Xinjiang Medical University (2022TSYCLJ0030 and 2021YFC2500605), and all participants signed informed consent forms.

## Data Availability

The raw data is available in the Supplementary File.

## Supplemental Information

Supplemental information for this article can be found online at http://dx.doi.org/10.7717/peerj.19344#supplemental-information.

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
