# Peer review of "Trends in dyslipidemia prevalence among Uyghur adults of different genders in China: a retrospective cohort study"

_PeerJ, doi:10.7717/peerj.19344_

## Round 0.1 · original submission · Minor Revisions

The reviewers point out some suggested modifications to this manuscript. Of note are the suggestions to provide increased detail regarding the study population (for example- adding non-HDL cholesterol values and commenting on findings, providing lipid profiles by age range, and additional information on cardiovascular status and smoking) and comparing results to similar populations in the discussion.

·

Basic reporting

I resd the manuscript entitled 'Trends in Dyslipidemia Prevalence Among Uyghur Adults of Different Genders in China: A Retrospective Cohort Study' with great interest. I think it deserves publication.

Strengths of the manuscript:
1. Significant Public Health Relevance: The study addresses the prevalence and trends of dyslipidemia, a critical CV risk factor, in a unique ethnic population (Uyghur adults). This population-specific focus enriches the global understanding of dyslipidemia in diverse genetic and cultural contexts.

2. Comprehensive Data Collection: A big sample size of 7,646 participants enhances the reliability of the findings. Longitudinal data spanning two years (2021–2023) allows for trend analysis, providing insights into the changing dynamics of lipid profiles and associated risk factors.

3. Ethnic and Gender-Specific Analysis: By analyzing the differences between genders and focusing on the Uyghur ethnic group, the study provides valuable insights into the interplay of genetic, lifestyle, and environmental factors on lipid metabolism.

4. Methodological Rigor: Standardized definitions based on the 2023 Chinese guidelines for lipid management ensure the study’s findings are aligned with current clinical standards.
The use of multivariable logistic regression to adjust for confounders strengthens the validity of the results.

5. Policy Implications: The results highlight specific areas for targeted interventions, such as obesity management and gender-sensitive strategies, which can be instrumental in formulating public health policies.

Limitations of the Manuscript

1. Demographic Representation: The study primarily includes rural participants from the Hotan region, potentially limiting the generalizability of findings to urban populations or other Uyghur communities in China.

2. Age Group Bias: Younger adults (18–39 years) are underrepresented, likely due to migration for education or work. This creates a gap in understanding dyslipidemia prevalence among this age group.

3. Lack of Dietary and Lifestyle Data: Although dietary habits and lifestyle factors (e.g., smoking, physical activity) are acknowledged as influencing lipid profiles, detailed data on these variables are not. presented, limiting a more nuanced analysis of causative factors.

4. Short Follow-Up Period: The two-year follow-up may be insufficient to capture the long-term trends and impacts of interventions on lipid profiles and cardiovascular outcomes.

5. Limited Urban-Rural Comparative Analysis: Including urban populations or conducting comparative analyses between rural and urban settings would provide a broader context for understanding dyslipidemia dynamics in the Uyghur population.

6. Biological and Genetic Investigations: While the study discusses genetic predispositions, no direct genetic analysis is conducted, which could provide deeper insights into the hereditary factors influencing dyslipidemia.

My Recommendations
1. Add non-HDL cholesterol values. Non-HDL= total cholesterol- HDL and comment on the findings
2. Please compare and discuss your results with the data from the same ethnic root i.e. Turkish populations. Data from Kazakhistan, Turkmenistan, an Turkey

Experimental design

Yes to all the 4 checklist

Validity of the findings

All to 3 checklist

This study offers valuable insights into the prevalence and trends of dyslipidemia among Uyghur adults and provides a foundation for future research and public health initiatives. While the strengths outweigh the limitations, addressing the identified gaps could further enhance the study's impact and applicability.

Reviewer 2 ·

Basic reporting

The study analyzed the prevalence and epidemiological characteristics of dyslipidemia among Uyghur adult population in China. It compared several parameters across gender, educational status, and metabolic profile in this population.

The manuscript was well written, short, and concise. Adequate references were used, and each section were well structured.

Most claims and conclusion made are consistent with existing literature. And the overall study addresses some knowledge gaps in this specific population.

Experimental design

No comment

Validity of the findings

- Since this study was limited to this population, more could be done by including result interpretation of lipid profile by Age ranges.

-Line 158 say: "Risk factors for dyslipidemia in males include higher educational attainment". This sounds a little ambiguous. Consider adding a context to this statement for simplicity.
Also consider adding a follow up sentence to the potential pathophysiology reason responsible for this.

Reviewer 3 ·

Basic reporting

This section of the manuscript is fine.
I could not review all supplementary files since many are in Chinese.

The text is clear and unambiguous, and professional English is used throughout.
Minor polishing of the language is required to fix typos and small errors.

Literature references are accurate and published in the last 5 years.
There is sufficient field background/context provided.

Self-contained with relevant results to hypotheses.

There is a professional article structure and original figures and tables. Raw data shared.
Table 2 - The prevalence of CHD remained the same for both men and women? Also the same SD?
Table 3 - you need to fix this table so that is has only 1 line for each sublipid fractions, it's difficult to follow the numbers
figure 1 - hypercholesterolemia - why is there such a high peak which then drops?

Experimental design

No comments
A statistician needs to verify the statistical analysis, I am not sufficiently skilled to do that

Validity of the findings

Why didn't you also analyze hypertension trends since you already had the data? Your population seems to include a lot of hypertensive patients. Smoking status should also be examined since it is also a cardiovascular risk factor.
The strengths and limitations section can be improved.
The discussion can also be strengthened with clinical recommendations, prevention strategies etc.

---

## Round 0.2 · accepted · Accept

Dear Dr. Maimaitituersun,

Thank you for submitting the revised version of your manuscript. After a thorough review of the changes by the reviewers and myself, I am pleased to inform you that all the reviewers' comments have been adequately addressed. Therefore, your manuscript is ready for publication in PeerJ.

Sincerely yours,

Stefano Menini

Reviewer 2 ·

Basic reporting

This retrospective cohort study looked at the trends in dyslipidemia prevalence among Uyghur adults.

Having looked at the manuscript on a second round, the authors have implemented all peer reviewers' comments, references well-aligned, and claims and conclusions are consistent with existing literature.

Experimental design

N/A

Validity of the findings

N/A

Reviewer 3 ·

Basic reporting

The authors have accurately addressed my comments and the paper can be accepted for publication

Experimental design

The authors have accurately addressed my comments and the paper can be accepted for publication

Validity of the findings

The authors have accurately addressed my comments and the paper can be accepted for publication

Additional comments

The authors have accurately addressed my comments and the paper can be accepted for publication